# Thoracic Inlet in Cervical Spine CT of Blunt Trauma Patients: Prevalence of Pathologies and Importance of CT Interpretation

Rathachai Kaewlai [1], Jitti Chatpuwaphat [1], Krittachat Butnian [1], Kittipott Thusneyapan [2], Nutthanun Panrong [2], Wanicha Lertpipopmetha [2] and Thongsak Wongpongsalee [3,*]

1 Division of Diagnostic Radiology, Department of Radiology, Faculty of Medicine Siriraj Hospital, Mahidol University, 2 Wanglang Rd., Bangkok Noi, Bangkok 10700, Thailand
2 Department of Anatomy, Faculty of Medicine Siriraj Hospital, Mahidol University, 2 Wanglang Rd., Bangkok Noi, Bangkok 10700, Thailand
3 Division of Trauma Surgery, Department of Surgery, Faculty of Medicine Siriraj Hospital, Mahidol University, 2 Wanglang Rd., Bangkok Noi, Bangkok 10700, Thailand
* Correspondence: wthongsak@gmail.com; Tel.: +66-86-015-5915

**Abstract:** Background: The thoracic inlet of blunt trauma patients may have pathologies that can be diagnosed on cervical spine computed tomography (CT) but that are not evident on concurrent portable chest radiography (pCXR). This retrospective investigation aimed to identify the prevalence of thoracic inlet pathologies on cervical spine CT and their importance by measuring the diagnostic performance of pCXR and the predictive factors of such abnormalities. Methods: This investigation was performed at a level-1 trauma center and included CT and concurrent pCXR of 385 consecutive adult patients (280 men, mean age of 47.6 years) who presented with suspected cervical spine injury. CT and pCXR findings were independently re-reviewed, and CT was considered the reference standard. Results: Traumatic, significant nontraumatic and nonsignificant pathologies were present at 23.4%, 23.6% and 58.2%, respectively. The most common traumatic diagnoses were pneumothorax (12.7%) and pulmonary contusion (10.4%). The most common significant nontraumatic findings were pulmonary nodules (8.1%), micronodules (6.8%) and septal thickening (4.2%). The prevalence of active tuberculosis was 3.4%. The sensitivity and positive predictive value of pCXR was 56.67% and 49.51% in diagnosing traumatic and 8.89% and 50% in significant nontraumatic pathologies. No demographic or pre-admission clinical factors could predict these abnormalities. Conclusions: Several significant pathologies of the thoracic inlet were visualized on trauma cervical spine CT. Since a concurrent pCXR was not sensitive and no demographic or clinical factors could predict these abnormalities, a liberal use of chest CT is suggested, particularly among those experiencing high-energy trauma with significant injuries of the thoracic inlet. If chest CT is not available, a meticulous evaluation of the thoracic inlet in the cervical spine CT of blunt trauma patients is important.

**Keywords:** adult; thoracic inlet; cervical vertebrae; tomography; X-ray computed; wounds; nonpenetrating

## 1. Introduction

Defined as the junction between the neck and the chest, the thoracic inlet is considered an "edge" of a series of images in the computed tomography (CT) of the cervical spine that can be easily overlooked [1]. The thoracic inlet contains several vital structures including lung apices, pleural spaces and superior mediastinum [1–3]. These structures may be injured in an acutely traumatized patient, resulting in significant pathologies such as pneumothorax, pneumomediastinum, mediastinal hematoma, pulmonary contusion and fractures [4]. Additionally, nontraumatic lesions incidentally found at the thoracic inlet may be clinically consequential, such as pulmonary nodules and active pulmonary infection.

Portable chest radiography (pCXR) is the current standard imaging performed during the adjunct to the primary survey according to the Advanced Trauma Life Support

(ATLS) protocol. The pCXR can reveal potentially life-threatening thoracic injuries such as pneumothorax and hemothorax that may escape clinical detection at the time of the initial assessment. Although the thoracic inlet is readily visualized on the trauma pCXR, the examination has a limited image quality, and abnormalities are potentially overlapped with bony structures and external objects [5]. Chest CT or whole-body CT (WBCT) is considered a valuable alternative for the evaluation of the thorax, inclusive of the thoracic inlet. However, it may not be performed in all trauma patients, as indications vary among different institutions [6,7].

Cervical spine CT is the recommended imaging in blunt-trauma patients suspected of having cervical spine injuries according to the ATLS protocol [8,9]. This examination typically covers the skull base to T1 or T2 vertebrae, and is therefore inclusive of many structures within the thoracic inlet, such as bones, pleurae and lung apices [1,10,11]. A few previous investigations have reported a high prevalence of incidental CT findings on trauma cervical spine CT [4,12,13]. Based on our experience, a careful review of thoracic inlet images included in a cervical spine CT of trauma patients can reveal potentially important findings not shown on a concurrent pCXR. These images are particularly helpful when chest CT is not concurrently performed.

Our objectives were to study the prevalence of thoracic inlet pathologies found on the cervical spine CT of blunt trauma patients and demonstrate their importance by evaluating the diagnostic performance of pCXR performed at the same time as CT, while also identifying factors potentially associated with pCXR abnormalities. This information will improve an understanding of the importance of identifying these pathologies on a trauma cervical spine CT.

## 2. Materials and Methods

### 2.1. Study Design and Patients

This retrospective single-center investigation was approved by the Human Research Protection Unit, Faculty of Medicine Siriraj Hospital, Mahidol University (protocol No. 874/2563 (IRB1) with COA No. Si 1019/2020), which considered it a minimal-risk study and waived the requirements for written informed consent. The study was performed in accordance with the relevant guidelines and regulations. The study setting was a level-I trauma center of a large urban academic hospital with a 2200-bed capacity. Between February 2020 and December 2020, all consecutive cervical spine CT scans of adult patients (age > 18 years) performed for blunt trauma mechanism at hospital admission were included. We excluded 169 scans because of various reasons detailed in Figure 1. The final study population was comprised of 385 patients, which met the required sample size based on the sensitivity of pCXR of 52% [14] with a 95% confidence interval and allowable error of 5%. At our institution, physicians use the National Emergency X-Radiography Utilization Study (NEXUS) criteria or Canadian Cervical Rule (CCR) to identify those requiring cervical spine clearance with imaging [15,16], where cervical spine CT is the imaging of choice.

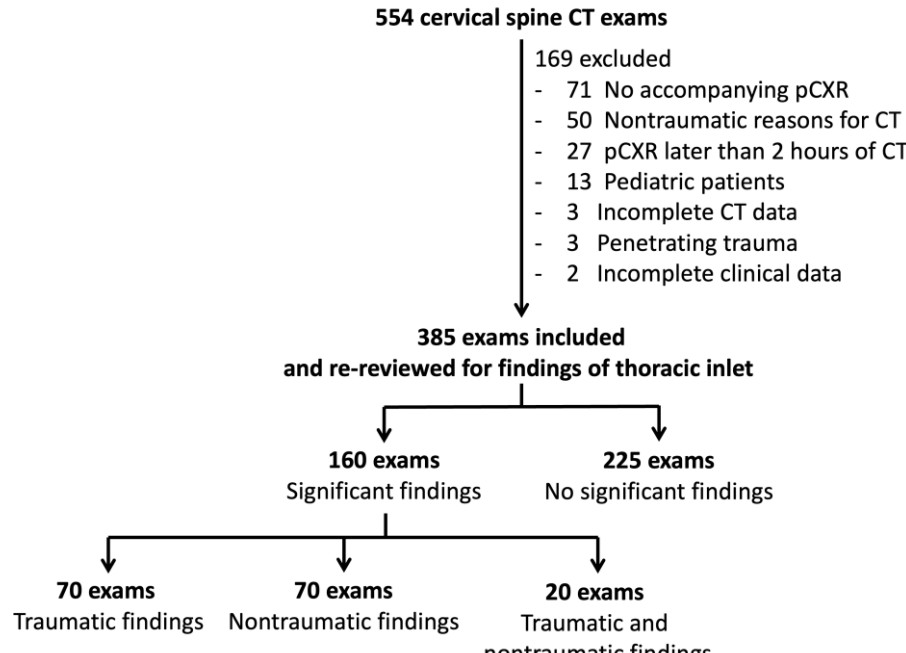

**Figure 1.** Flowchart of patient inclusion.

### 2.2. Image Acquisition

All cervical spine CT scans were performed either on a 64-slice MDCT (Discovery CT750 HD, GE Healthcare, Chicago, IL, USA) or a 256-slice MDCT (Revolution CT, GE Healthcare, Chicago, IL, USA) without intravenous contrast administration. Scan parameters were set to 120 kVp, 150–400 mA with an automatic tube current modulation, rotation time of 0.6 s, pitch of 0.984:1 mm/rotation and noise index of 7 (64-slice MDCT), or 120 kVp, 165–375 mA with an automatic tube current modulation, rotation time of 0.8 s, pitch of 0.984:1 mm/rotation and noise index of 10 (256-slice MDCT). Each scan was obtained with an axial slice thickness of 0.625 mm, covering the vertex (for a head-including cervical spine exam) or skull base (for a cervical spine exam) to the lower edge of T2 using a detector coverage of 40 cm. Axial images (1.25-mm slice thickness) sent to the Picture Archiving and Communication Systems (PACS; Synapse, Fujifilm Corporation, Tokyo, Japan) included soft-tissue and bone algorithms. All pCXR examinations were performed on a portable machine (AccE GM85, Samsung Healthcare, Seoul, Korea) with 85 kVp, 320 mA and 5 ms, with patients in a supine position.

### 2.3. Original Reports

Original radiology reports were interpreted and finalized by a group of radiologists (n = 20) with a median of 8.5 years of experience (range, 1–33) after completion of radiology residency. Fourteen practiced neuroimaging, while the others consisted of body imagers (n = 4), emergency imagers (n = 1) and an interventional neuroradiologist (n = 1). Twelve were attending radiologists, and the others were trainees in their first or second year of neuroradiology fellowship training. The findings of thoracic inlets in original radiology reports were extracted and categorized into those with traumatic (n = 26), significant nontraumatic (n = 84) and without any significant findings (n = 275).

### 2.4. Image Re-Interpretation

One body and one thoracic subspecialty radiologist, both with 5 years of experience after completion of radiology residency (3 years after fellowship completion), independently re-reviewed the cervical spine CT and pCXR of all patients on a PACS workstation with the ability to adjust the window level/width and (for CT) image orientation. Image re-review was performed in two separate sessions. First, pCXR scans of all patients were reviewed,

and thoracic inlet findings were documented. Second, the CT of all patients (provided in a different order list) were reviewed. In each session, the reviewers were blind to the findings in the other accompanying imaging studies. For the pCXR review, radiologists were informed to look specifically at the thoracic inlet to identify any abnormalities. If abnormalities were detected, they provided details and characterization. For the CT review, reviewers assessed images below the C7 vertebral body on an axial plane in soft tissue, bone and lung windows. They were able to perform additional coronal and sagittal reformats using the same PACS software at their discretion.

The radiologists were blinded to clinical data except for a history of blunt trauma. All disagreements between two radiologists were resolved by a third radiologist who subspecialized in emergency radiology with 20 years of experience.

### 2.5. Definitions, Categories, Appearances of Findings and Reference Standard

Definitions of most data were self-explanatory. Low-energy trauma included fall from standing and assaults. All motorcycle collisions were considered high energy. "Other" mechanisms of trauma included hanging and drowning. The Glasgow Coma Scale (GCS) score, Abbreviated Injury Score (AIS) and Injury Severity Score (ISS) were calculated based on established standards. The presence of cervical spine injuries was detected on CT and confirmed as discharge diagnoses in the patient's medical records. Presentations outside of normal working hours were those between 4:00 p.m. and 8:00 a.m. of the next day on weekdays, and around the clock on weekends.

The definitions of each imaging finding and the four categories of these findings are provided in Table S1 [1,10,11]. Findings indicative of injury were classified as trauma-related, and those not indicative of injury were classified as not trauma-related. CT findings were considered a reference standard for all pathologies examined in this investigation. For those with ground-glass opacities and CT findings suggestive of pulmonary contusion, laboratory test (real-time polymerase chain reaction; RT-PCR) results for coronavirus disease 2019 (COVID-19) were assessed and reported.

### 2.6. Statistical Analysis

Categorical variables (such as gender, trauma mechanism, AIS, CT and pCXR findings) were presented as a number or percentage. Continuous data (such as age, time lapse from trauma onset, length of stay) were reported as a mean (standard deviation) or median (range) depending on the distribution of the data. The performance of pCXR was derived from a 2 × 2 table and reported as sensitivity, specificity, accuracy, positive predictive value (PPV), negative predictive value (NPV), positive likelihood ratio (PLR) and negative likelihood ratio (NLR) with 95% confidence intervals. The Statistical Package for Social Sciences (SPSS, version 23, IBM, Chicago, IL, USA) was utilized for these analyses.

## 3. Results

### 3.1. Patient and Study Characteristics

There was a total of 385 patients (with a similar number of CT exams) included in this investigation (Table 1). Of these, 280 were men (280/385; 72.7%) with a mean age of 47.6 years (SD 22.1). Twenty-six patients (26/385; 6.8%) had cervical spine injuries. The median ISS was 10 (range 0–45; IQR 14).

**Table 1.** Comparison among patients with significant findings (Group 1; trauma, significant nontraumatic, both traumatic and nontraumatic findings) and those without (Group 2) *.

| | | Group 1 | | | Group 2 | |
| --- | --- | --- | --- | --- | --- | --- |
| | **All Patients (n = 385)** | **1.1 Trauma (n = 70)** | **1.2 Significant Nontraumatic (n = 70)** | **1.3 Trauma and Significant Nontraumatic (n = 20)** | **2. No Trauma or Significant Nontraumatic (n = 225)** | ***p*-Value between Group 1 and Group 2 ** |
| Demographics & clinical data | | | | | | |
| Male gender | 280 | 56 (80) | 48 (68.6) | 15 (75) | 161 (71.6) | 0.620 |
| Age (years; mean, SD) | 47.6 (22.1) | 37.6 (17.7) | 55.3 (21.0) | 54.9 (26.6) | 47.7 (22.2) | 0.902 |
| Trauma mechanism | | | | | | 0.374 |
| Fall from standing | 102 | 2 | 29 | 6 | 65 | |
| Fall from height | 22 | 4 | 2 | 1 | 15 | |
| Motorcycle collision | 177 | 50 | 21 | 6 | 100 | |
| Car collision | 17 | 7 | 1 | 1 | 8 | |
| Pedestrian/ bike accident | 27 | 3 | 7 | 2 | 15 | |
| Assault | 17 | - | 4 | 1 | 12 | |
| Others | 23 | 4 | 6 | 3 | 10 | |
| High-energy trauma mechanism (n = 362) | 243 | 64 | 31 | 10 | 138 | 0.185 |
| Time lapse from trauma onset (hours; median, range, IQR) | 1 (1–336, 2) | 1 (1–72, 1) | 1 (1–37, 2) | 1 (1–336, 3) | 1 (1–168, 2) | 0.623 |
| GCS groups | | | | | | 0.149 |
| Full (GCS = 15) | 190 | 32 | 35 | 4 | 119 | |
| Mild (GCS = 13–14) | 86 | 15 | 15 | 6 | 50 | |
| Moderate (GCS = 9–12) | 48 | 6 | 10 | 4 | 28 | |
| Severe (GCS = 3–8) | 61 | 17 | 10 | 6 | 28 | |
| Presence of C-spine injuries | 26 | 6 | 3 | 14 | 3 | 0.817 |
| AIS: Head (median, range, IQR) | 2 (0–5, 2) | 2 (0–5, 1) | 2 (0–5, 1) | 2 (0–5, 1) | 2 (0–5, 2) | 0.068 |
| AIS: Face (median, range, IQR) | 0 (0–4, 1) | 0 (0–4, 2) | 0 (0–3, 1) | 0 (0–4, 1) | 0 (0–4, 1.5) | 0.065 |
| AIS: Thorax (median, range, IQR) | 0 (0–5, 0) | 1 (0–5, 3) | 0 (0–3, 0) | 0 (0–4, 2) | 0 (0–3, 0) | 0.000 |
| AIS: Abdomen (median, range, IQR) | 0 (0–5, 0) | 0 (0–5, 0) | 0 (0–3, 0) | 0 (0–2, 0) | 0 (0–5, 0) | 0.010 |
| AIS: Extremity (median, range, IQR) | 0 (0–4, 1) | 0 (0–3, 2) | 0 (0–3, 1) | 0 (0–4, 2) | 0 (0–4, 1) | 0.039 |
| AIS: External (median, range, IQR) | 0 (0–5, 0) | 0 (0–5, 0) | 1 (0–3, 0) | 1 (0–1, 1) | 1 (0–2, 0) | 0.130 |
| ISS (median, range, IQR) | 10 (0–45, 14) | 18 (1–45, 20) | 10 (1–34, 12) | 11 (1–36, 19) | 9 (0–41, 12) | 0.000 |

**Table 1.** *Cont.*

| | All Patients (n = 385) | 1.1 Trauma (n = 70) | Group 1 | | Group 2 | *p*-Value between Group 1 and Group 2 ** |
| | | | 1.2 Significant Nontraumatic (n = 70) | 1.3 Trauma and Significant Nontraumatic (n = 20) | 2. No Trauma or Significant Nontraumatic (n = 225) | |
|---|---|---|---|---|---|---|
| Presentation outside of normal working hours | 276 | 52 | 52 | 9 | 163 | 0.783 |
| Treatment of neck injuries | | | | | | 0.479 |
| None | 225 | 31 | 48 | 14 | 132 | |
| Cervical collar | 158 | 39 | 22 | 6 | 91 | |
| Surgery | 2 | 0 | 0 | 0 | 2 | |
| Treatment of chest injuries | | | | | | 0.000 |
| None | 361 | 53 | 69 | 18 | 221 | |
| Unilateral or bilateral ICD | 23 | 16 | 1 | 2 | 4 | |
| TEVAR | 1 | 1 | 0 | 0 | 0 | |
| Hospital admission | 196 | 42 | 37 | 13 | 104 | 0.038 |
| Length of stay (days; median, range, IQR; n = 196) | 6 (0–180, 11) | 10 (0–92, 15) | 4 (0–27, 8) | 8 (0–34, 10) | 5 (0–180, 9) | 0.434 |
| Discharge status (n = 364) | | | | | | 0.056 |
| Death | 20 | 4 | 6 | 3 | 7 | |
| Transfer | 107 | 24 | 18 | 5 | 60 | |
| Alive | 237 | 39 | 41 | 11 | 146 | |
| Length of follow up (days; median, range, IQR) | 21 (0–540, 194) | 29 (0–474, 185) | 9 (0–532, 290) | 4 (0–287, 96) | 22 (0–540, 200) | 0.915 |
| CT imaging data | | | | | | |
| C-spine CT including head CT | 360 | 67 | 65 | 20 | 208 | 0.428 |
| Chest CT within 24 h of trauma | 31 | 18 | 3 | 4 | 6 | 0.0001 |
| Official CT report verified by attending radiologist ** | 210 | 37 | 46 | 9 | 118 | 0.380 |
| Reporting radiologist experience >10 years | 183 | 32 | 40 | 8 | 103 | 0.475 |

AIS = abbreviated injury score, GCS = Glasgow Coma Score, IQR = interquartile range, ISS = injury severity score, TEVAR = thoracic endovascular aortic repair. Number of patients stated at column headers unless specified otherwise within a row header. * Categorical and continuous variables are compared among two groups using independent *t*-test or Mann-Whitney U test, respectively. *p*-values are considered significant when below 0.05. ** versus those verified by neuroradiology fellows.

The original CT reports were interpreted by an attending radiologist (210/385; 54.5%), while the rest were done by neuroradiology fellows. Of all 385 reports, 362 (94%) were finalized by those in the neuroimaging subspecialty. One hundred and eighty-three reports (183/385; 47.5%) were verified by radiologists with 10 years of experience or more.

### 3.2. Prevalence and Details of Pathologies of Thoracic Inlet

Traumatic, significant nontraumatic and nonsignificant pathologies of the thoracic inlet (Table 2) were found in 92, 90 and 224 patients, which accounted for a prevalence of 23.90%, 23.56% and 58.18%, respectively. The two most frequent traumatic pathologies were pneumothorax (49/385; 12.7%) and pulmonary contusion (40/385; 10.4%). Among patients with pulmonary contusion, five were tested and all were negative for COVID-19.

**Table 2.** Prevalence of thoracic inlet pathologies diagnosed with C-spine CT and their corresponding chest radiographic findings.

| Thoracic Inlet Pathologies | N (%) | Corresponding Radiographic Findings |
|---|---|---|
| **Traumatic findings** | **92 (23.90)** | |
| Mediastinal fat stranding | 5 | Abnormal mediastinum |
| Mediastinal hemorrhage | 4 | Abnormal mediastinum |
| Pneumomediastinum | 1 | Pneumomediastinum |
| Pulmonary contusion | 40 | Patchy opacification |
| Pulmonary laceration | 3 | None |
| Pneumothorax | 49 | Pneumothorax |
| Pleural fluid | 9 | Apical cap |
| Extrapleural hematoma | 2 | Apical cap |
| Rib fracture: first | 13 | Rib fracture: first |
| Rib fracture: second | 14 | Rib fracture: second |
| Rib fracture: third | 9 | Rib fracture: third |
| Rib fracture: fourth | 7 | Rib fracture: fourth |
| Clavicle fracture | 8 | Clavicle fracture |
| Acromioclavicular dislocation | 0 | Acromioclavicular dislocation |
| Scapular fracture | 1 | Scapular fracture |
| **Significant nontraumatic** | **90 (23.38)** | |
| Mediastinal vascular dilation | 4 | Abnormal mediastinum |
| Pulmonary nodule(s) | 31 | Pulmonary nodule(s) |
| Pulmonary micronodules | 26 | N/A |
| Groundglass opacity | 6 | N/A |
| Groundglass nodule(s) | 5 | N/A |
| Cavity | 3 | Cavity |
| Atelectasis | 2 | Increased opacity with volume loss |
| Septal thickening * | 24 | N/A |
| Active tuberculosis | 13 | Active tuberculosis |
| Pulmonary malignancy | 2 | Pulmonary malignancy |
| Foreign body | 1 | Radiopaque foreign body |
| Malignant bone lesions | 1 | Lucent bone lesions |

**Table 2.** *Cont.*

| Thoracic Inlet Pathologies | N (%) | Corresponding Radiographic Findings |
|---|---|---|
| **Non-significant abnormalities** | **224 (58.18)** | |
| Parenchymal scars | 74 | Parenchymal scars |
| Calcifications | 15 | Calcifications |
| Bronchiectasis | 24 | Bronchiectasis |
| Emphysema | 62 | Emphysema |
| Blebs/bulla | 55 | Blebs/bulla |
| Benign bone lesions | 4 | Lucent bone lesions |

\* Excluding those with pulmonary contusion.

For significant nontraumatic pathologies, the three most frequent findings included pulmonary nodule(s), micronodules and septal thickening at rates of 8.1% (31/385), 6.8% (26/385) and 6.2% (24/385), respectively. The three most common nonsignificant pathologies were parenchymal scars, emphysema and blebs/bulla, which had a prevalence of 19.2% (74/385), 16.1% (62/385) and 14.3% (55/385), respectively.

*3.3. Performance of Portable CXR and Original CT Reports*

The pCXR had a 36.15% sensitivity, 94.82% specificity, 56.77% accuracy, 92.78% PPV, 44.59% NPV, 6.97 PLR and 0.67 NLR for the detection of "any" findings of thoracic inlet (Table S2). For traumatic findings, pCXR had an overall sensitivity of 56.67%, overall specificity of 82.31% and NPV of 86.12. The overall pCXR sensitivity and PPV for significant nontraumatic findings were 8.89% and 50%, respectively.

Higher AIS-Thorax, AIS-Abdomen, AIS-Extremity, ISS, treatment of chest injuries, hospital admission and chest CT performance within 24 h of cervical spine CT were associated with having traumatic or significant nontraumatic findings of thoracic inlet on univariate analysis. However, none were significant on multivariate analysis.

The original C-spine CT reports had low sensitivities (27.17–40.37%) but modest specificities (79.91–99.66%) for the identification of traumatic, significant nontraumatic and both findings (Table 3).

**Table 3.** Performance of original C-spine CT report in the identification of thoracic inlet abnormalities using re-interpretation as a reference standard.

| | True Positive | False Positive | False Negative | True Negative | Sensitivity (95% CI) | Specificity (95% CI) | Accuracy (95% CI) |
|---|---|---|---|---|---|---|---|
| **Overall performance** | 65 | 45 | 96 | 179 | 40.37 (32.72–48.38) | 79.91 (74.06–84.95) | 63.38 (58.35–68.20) |
| **All traumatic findings** | 25 | 1 | 67 | 292 | 27.17 (18.42–37.45) | 99.66 (98.11–99.99) | 82.34 (78.15–86.02) |
| **All significant nontraumatic findings** | 8 | 12 | 13 | 59 | 38.10 (18.11–61.56) | 83.10 (72.34–90.95) | 72.83 (62.55–81.58) |

## 4. Discussion

Our investigation reveals a much higher prevalence of incidental findings in trauma cervical spine CT scans than reported in previous studies, with rates ranging from 23.44% to 58.18%. Barboza et al. [4] assessed 1256 cervical spine CTs, which revealed incidental pulmonary and mediastinal findings in 84 cases (6.7%), while Paluska et al. [17] demonstrated a 15.9% prevalence of apical nontraumatic thoracic findings among 289 trauma cervical spine CTs. In the former study [4], traumatic lesions of thoracic inlet included

rib fracture (3.2%), pneumothorax (1.7%), contusion (1.5%), pneumomediastinum (0.3%), effusion (0.2%), clavicle fracture (0.1%) and manubrial fracture (0.1%). For nontraumatic findings, the latter study [17] revealed lung nodules (2.8%), bulla (1%), emphysema (1%), blebs (0.7%), granulomatous disease (0.7%), scarring (0.3%) and lung mass (0.3%). The prevalence of abnormalities of thoracic inlet in our investigation is much higher than in prior studies [4]. The discrepant prevalence may stem from a high proportion of patients sustaining high-energy trauma in our cohort, making them more likely to have injuries of the thoracic inlet in addition to the cervical spine. Multiple other factors encompass scan techniques (i.e., thinner slice thickness in our investigations; caudal coverage of scans), methods used for interpretation (i.e., ability to perform/use multiplanar reformations), the subspecialty of radiologists who re-interpret CT scans, and specified research questions.

We also found discrepancies between re-interpretation and original radiology reports, in which both under- and overreads exist. Beheshtian et al. [13] found that only 20 out of 701 incidentalomas (2.9%) were documented in 2116 original cervical spine CT reports. In contrast, Barboza et al. [4] identified a 9.1% missed rate in 1256 official trauma cervical spine CT reports, in which the majority (75%) of incidentalomas were nontraumatic findings. We illustrated common findings in our cohort in Figures S1–S6 using recommended viewing standards of a less-than-3-mm slice thickness, multiplanar reformations, multiple window settings, and soft-tissue and lung algorithms [18].

Because COVID-19 pneumonia has overlapping CT findings with pulmonary contusions, it can pose as a mimicker that might falsely increase the prevalence of this injury. A recent investigation has revealed a high prevalence (70.5%) of COVID-19 pneumonia in apical lungs of trauma patients having a cervical spine CT [19]. We did not find any positive COVID-19 cases in our cohort, but the number that was tested for was small. The low rate of COVID-19 testing likely reflected the status of this infection at the time of this research, when there were only 6884 confirmed COVID-19 cases in Thailand [20]. This, therefore, does not exclude a possibility of COVID-19 pneumonia being wrongly assigned as pulmonary contusion in our cohort.

Because most of our patients (90.1%) did not undergo chest CT, pCXR performed concurrently with cervical spine CT becomes the de facto imaging to rely upon for thoracic inlet findings. We examined the diagnostic performance of pCXR in comparison with cervical spine CT and found that pCXR was poorly sensitive for significant traumatic and nontraumatic findings of the thoracic inlet. This occurred even though pCXRs were re-reviewed by subspecialists. This is not unexpected, as pCXR has multiple limitations that can be pathology-, patient- or technique-related [9]. Our investigation reaffirms the notion of the limited performance of pCXR in diagnosing significant findings of thoracic inlet and suggests that better diagnostic tools, such as chest or whole-body CT, should be used more liberally, particularly in those experiencing high-energy trauma with significant traumatic injuries at the thoracic inlet. Whether one should avoid pCXR entirely and shift an imaging workup to CT is a matter of debate, as CT comes at a significant financial cost, may produce incidental findings and results in radiation burden, potentially increasing the risk of future malignancy, especially in young patients [21]. pCXR is still considered a helpful adjunct to the primary survey of the ATLS protocol [9] for detecting potentially life-threatening injuries and can be taken during resuscitation.

Although several factors were significantly different between groups having versus not having traumatic and significant nontraumatic findings of thoracic inlet, none were present independently from others, nor could any be determined before admission. Others have reported that female gender and older age are associated with an increased probability of nontraumatic incidentalomas [17]. Older age, a high-energy mechanism of injury and higher ISS [4] have been associated with a greater proportion of incidental findings. We also identified AIS-Thorax, AIS-Abdomen, AIS-Extremity, ISS, treatment of chest injuries, hospital admission and chest CT performance within 24 h of cervical spine CT as predictive of thoracic inlet abnormalities, but these were neither independent predictors nor present at the time of the initial decision-making. Interestingly, we did not identify high-energy

trauma as an independent predictor, but Kelleher et al. [22] suggested a link between the severity of the trauma mechanism and the presence of injuries on chest CT. In their study of 115 patients sustaining low-energy trauma with suspected head/cervical spine injury, none had positive traumatic findings on accompanying chest/abdomen/pelvis CT.

Our investigation is limited by its retrospective nature. Cases were acquired through a retrospective search instead of a prospectively maintained trauma registry, as the latter is not available at our institution. This may introduce a selection bias as cases may unintentionally be excluded or missed. Although the sample size is relatively small, it reached the precalculated level for ensuring adequate statistical power in determining the diagnostic performance of concurrent pCXR. We chose to rely on CT as the reference standard for diagnosis, as it is well known that CT has great sensitivity in detecting thoracic injuries [23,24], although certain features (i.e., vascular, soft tissue abnormalities) may be difficult to prove without the administration of intravenous contrast medium. We are aware that CT findings do not necessarily translate to intervention, although certain findings may require a follow-up. Aside from clinical follow-ups, we did not assess for repeat imaging of these findings, as this is out of our research scope. However, we believe that our imaging follow-up rates would likely be small, as there are no clinical protocols for follow-ups in our institution, and the rates were reportedly generally suboptimal [17,25,26]. Although we emphasize the importance of looking at thoracic inlet while interpreting a cervical spine CT, its value may diminish and cannot be generalized if chest CT is commonly performed concurrently with cervical spine CT or as a whole-body CT. This was not the case in our institute, as chest CT was not frequently performed. As a result, we did not exclude patients with chest CT that was performed close to, or concurrently with, cervical spine CT, as this would unnecessarily introduce a selection bias. We did not evaluate scout CT images, as this is out of our scope; however, these images have been shown to reveal many potentially important abnormalities of the thorax [27].

In conclusion, significant thoracic inlet findings on the cervical spine CTs of blunt trauma patients, particularly in those who sustained a high-energy mechanism, were common and frequently undiagnosed on a concurrent pCXR. As no pre-admission clinical factors were able to predict such abnormalities, it becomes clear that chest CT should be used more liberally in this patient group. If it is not available, the responsibility for injury identification will fall upon the radiologist who interprets a cervical spine CT. Such a role requires the careful assessment of the structures of the thoracic inlet, including the lung apices, pleura, mediastinum and bones.

**Supplementary Materials:** The following supporting information can be downloaded at: https://www.mdpi.com/article/10.3390/tomography8060231/s1, Figure S1: pneumothorax and subpleural bleb/bulla; Figure S2: pulmonary contusion; Figure S3: mediastinal hemorrhage; Figure S4: rib and clavicle fractures; Figure S5: subsolid pulmonary nodules; Figure S6: pulmonary micronodules and cavities; Table S1: CT findings of thoracic inlet pathologies and their corresponding chest radiographic findings; Table S2: performance of portable trauma chest radiography in identification of thoracic inlet abnormalities.

**Author Contributions:** Conceptualization, R.K. and T.W.; methodology, R.K. and T.W.; software, R.K. and T.W.; validation, R.K.; formal analysis, R.K.; investigation, R.K., J.C., K.B., K.T., N.P., W.L. and T.W.; resources, R.K. and T.W.; data curation, R.K.; writing—original draft preparation, R.K.; writing—review and editing, R.K., J.C., K.B., K.T., N.P., W.L. and T.W.; visualization, R.K.; supervision, R.K. and T.W.; project administration, R.K. and T.W.; funding acquisition, N/A. All authors have read and agreed to the published version of the manuscript.

**Funding:** This research received no external funding.

**Institutional Review Board Statement:** The study was conducted in accordance with the Declaration of Helsinki and approved by the Institutional Review Board of Faculty of Medicine Siriraj Hospital, Mahidol University (protocol code 874/2563 on 11 December 2020).

**Informed Consent Statement:** Patient consent was waived due to minimal risk.

**Data Availability Statement:** Datasets generated and/or analyzed during the current study are not publicly available due to their nature as personal health information but may be available from the corresponding author on reasonable request.

**Acknowledgments:** We would like to thank Sasima Tongsai, Clinical Epidemiology Unit, Research Department, Faculty of Medicine Siriraj Hospital, for her help with statistical analysis.

**Conflicts of Interest:** The authors declare no conflict of interest.

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
