# Peer review of "Thoracic Inlet in Cervical Spine CT of Blunt Trauma Patients: Prevalence of Pathologies and Importance of CT Interpretation"

_tomography, doi:10.3390/tomography8060231_

Round 1
Reviewer 1 Report
Dear Authors,
thank you for submitting this interesting article about the CT interpretation of the thoracic inlet of blunt trauma patients. The paper is well designed with a significative statistical analysis.
I would suggest clarifying the purpose of your work adding a more extensive explanation of the CXR role in blunt trauma patients in the background section.
Moreover, you affirm that CT should be used more liberally, particularly in those experiencing high-energy trauma with significant traumatic injuries at the thoracic inlet. I would just suggest to better highlight this message, which is just mentioned in the discussion, considering the notion reaffirmed by your results of CXR limited performance.
Best regards
Author Response
Point 1: I would suggest clarifying the purpose of your work adding a more extensive explanation of the CXR role in blunt trauma patients in the background section.
Response 1: We agree with the reviewer. We have added details, rearranged the paragraphs and references in the Introduction of the revision (Line 48-62).
Point 2: Moreover, you affirm that CT should be used more liberally, particularly in those experiencing high-energy trauma with significant traumatic injuries at the thoracic inlet. I would just suggest to better highlight this message, which is just mentioned in the discussion, considering the notion reaffirmed by your results of CXR limited performance.
Response 2: We agree with the reviewer and have added a sentence to the Conclusion (Line 30-33) and the end of Discussion (Line 304-308) in the revised manuscript.
Reviewer 2 Report
1. I would like to know how the authors cope with the annotation shift results from different annotator experiences and the selection bias caused by the data curating process.
2. The authors mentioned that "COVID-19 pneumonia has overlapping CT findings with pulmonary contusions" and is there any possibility of avoiding the spurious correlation?
3. Does patient distribution affect the results?
Author Response
Point 1: I would like to know how the authors cope with the annotation shift results from different annotator experiences and the selection bias caused by the data curating process.
Response 1: We agree that radiologists of different experiences may annotate or identify lesions differently. Therefore, we have tried to unify the re-interpretation process by having three radiologists where the two independently re-interpreted all studies and the third radiologist adjudicated on all disagreements. Since CT findings of the thoracic inlet are self-explanatory for the radiologists and do not include severity assessment of the pathologies, we did not provide image examples.
The data curated for our investigation are derived from retrospective analysis, including all clinical information, instead of a prospectively maintained trauma registry database. This has a potential for selection bias as cases may unintentionally be missed or excluded. We have added this to the limitation in the revision (Line 281-284).
Point 2: The authors mentioned that "COVID-19 pneumonia has overlapping CT findings with pulmonary contusions" and is there any possibility of avoiding the spurious correlation?
Response 2: We agree with the reviewer. Since not all patients with pulmonary contusion were tested for COVID-19, it was difficult or impossible to exclude COVID-19 as an alternative explanation for abnormal pulmonary opacities in our cohort. We have added this as a limitation of the paper (Line 245-250).
Point 3: Does patient distribution affect the results?
Response 3: We agree with the reviewer. A high prevalence of high-energy trauma in our cohort might have affected the results (high prevalence of thoracic inlet findings). We have added this to the discussion in the Revision (Line 225-228).